# An Educational Intervention for Improving the Snacks and Beverages Brought to Youth Sports in the USA

**DOI:** 10.3390/ijerph18094886

**Published:** 2021-05-04

**Authors:** Lori Andersen Spruance, Natalie Bennion, Gabriel Ghanadan, Jay E. Maddock

**Affiliations:** 1Department of Public Health, Brigham Young University, Provo, UT 84602, USA; nataliebennion@gmail.com (N.B.); bhgabe@gmail.com (G.G.); 2Department of Environmental and Occupational Health, Texas A&M University, College Station, TX 77843, USA; maddock@tamu.edu

**Keywords:** youth, sports, sugar-sweetened beverages, water

## Abstract

Objectives: The purpose of this study was to test a small-scale intervention and its ability to decrease total sugar intake and number of calories offered at youth sports games. Methods: This study was a pre/post-test quasi-experimental design. A flier was developed and distributed to parents. The flier aimed to decrease the sugar-sweetened beverages and increase the nutritional quality of food brought to games. Baseline data were collected in 2018 (*n* = 61). The flier was distributed prior to the start of the league, once during the league, and posted online in 2019. Postintervention data were collected in the intervention group (*n* = 122) and a comparison group (*n* = 74). Nutritional information was collected through direct observation. Results: The average amount of total sugar provided per game per child was 25.5 g at baseline when snacks/beverages were provided at games. After the intervention, the average amount of total sugar provided significantly decreased (16.7 g/game/child, *p* < 0.001). Conclusions: The intervention reduced total sugar offered and the number of sugar-sweetened beverages brought to games. It was low-cost and could be easily implemented by public health practitioners and/or parks and recreation administrators. Further, considerations could be made to implement policies relative to snacks and beverages at youth sports games.

## 1. Introduction

Poor eating habits are one of the largest factors contributing to the development of chronic disease. High consumption of simple carbohydrates, such as added sugars, has been correlated with chronic conditions such as obesity, heart disease, high blood pressure, and fatty liver disease [1]. Establishing healthy eating habits is especially important among children because many dietary habits carry from childhood into adulthood [2].

The percentage of school-aged children with obesity in 2016 was almost 20% and is highest among Non-Hispanic Black and Hispanic children in the USA [3]. Childhood obesity increases over the last few decades may be attributed to recent changes in children’s dietary and physical activity habits [4]. Research has determined that snacks alone account for over 460 calories of a child’s daily diet and almost 15% of their food and beverage consumption comes from added sugar [5].

The Social–Ecological Model (SEM) highlights the complex interplay of several systems, including individual, interpersonal, community, and societal factors [6]. The Academy of Nutrition and Dietetics developed a specific SEM to guide understanding of dietary behaviors [7]. Children engage with multiple environments that influence their diet and exercise behaviors [7]. Youth sports is one environment that can influence dietary behaviors [7]. 

Several studies have found that youth recreational sports have an effect on the overall diet in children. In the USA, recreational sports typically do not require individuals to try out for teams and focus on enjoyment and development versus travel or competition. For example, one study found that 62% of families go out to eat after recreational soccer games [8] and food consumed outside the home and school settings is associated with a greater intake of noncore foods and increased caloric intake [9,10]. Two cross-sectional dissertation studies found that snacks brought to recreational soccer games by parents in the USA usually have between 300 and 500 calories, and snacks are typically grain-based desserts [11,12]. In US diets, grain-based desserts are the second largest source of food that contribute to added sugars [13]. Another study identified a relationship between high physical activity levels and high sugar-sweetened beverage (SSB) consumption among 6–11-year-old children, likely related to SSB consumption offered as part of youth sport participation [14]. Additionally, Bennion et al. discovered that children consume more calories than they expend during youth recreational sports [15]. In another qualitative study, parents indicated that team snacks are a large part of the reason their children play [16]. Parent participants described that unhealthy snacks are common at games because they are inexpensive, easy to prepare, and appealing to children [16,17]. In other studies, parent participants indicated they were interested in receiving specific nutritional guidelines for recreational sports [17,18].

There are a limited number of interventions that focus on changing the dietary behaviors of children participating in youth recreational sports. Some studies suggested that local parks and recreation departments increase education efforts towards parents on maintaining healthy eating behaviors [8,17]. One study found that healthy snacks were more likely to be sold in sports facilities when signs were used to advertise, taste testing was offered, and price was discounted by 30% [19]. Another study applied traffic light labels to snacks at a recreational facility and found that healthy products marked with green lights increased in sales and unhealthy products marked with red lights decreased in sales, while simultaneously maintaining daily revenue for the sports facility [20]. However, these intervention strategies specifically focused on changing food for sale at youth sport facilities, rather than the snacks and beverages actually offered during or after games by parents/guardians. Because few interventions have explicitly targeted the snacks and beverages brought to youth sports, effective interventions are needed.

### Purpose

The purpose of this study was to test a small-scale educational intervention and its ability to improve the youth sports food environment among the types of snacks/beverages brought to games by parents and their nutritional content. Our main area purpose was to decrease the total calories offered at each game and reduce the amount of sugar offered at each game. Secondary aims of the study sought to quantify the other nutrients offered including sodium, fiber, cholesterol, protein, fat, and saturated fat. The hypothesis of the study was that after the intervention, the number of calories offered would decrease, the amount of sugar offered would decrease, and the amount of fruits, vegetables, and water brought to games would increase.

## 2. Methods

### 2.1. Study Design and Participants

This quasi-experimental study was conducted among 3rd and 4th grade children participating in soccer within two parks and recreational leagues in Utah, USA. Teams were organized by the parks and recreation departments and remained together for the entire season. Baseline observations (*n* = 61 games observed) occurred in April 2018, and postintervention observations (*n* = 122 games observed) occurred in April 2019. Because data were collected observing the same age-group league, it is likely that these are not the same parents and children given maturation and nonparticipation in sport from one year to the next. Baseline data were previously published [15]. One parks and recreational league received an intervention targeting the snacks and beverages parents bring to games, while the other league served as a comparison group. Comparison data (*n* = 74) were collected from the second league after the intervention was accidentally sent to all baseline participants by the parks and recreation department prior to data collection; baseline data were not collected for the comparison group. Data from this league were collected in April 2019. In these particular jurisdictions, there were no concession stands or outside vendors at the sports grounds.

Games were randomized for observation. A total of 120 games were scheduled for the intervention group, and 99 games were scheduled for the comparison group. A minimum of 50% of the games for both intervention and comparison were selected for observation. In the event that the games were canceled due to weather, additional observations were added from the randomized list to offset the canceled games. Baseline data were collected as part of a previous study. Two research assistants attended each game and observed opposing teams, thus two observations occurred at each scheduled game.

Trainings were conducted and observation inter-rater reliability was established prior to the start of data collection. Research assistants were required to independently observe the snacks and beverages served during or after the game to one team and compare their results to the other observer. If the pair of observers obtained a reliability score of 80% or higher, they were cleared to perform observations for data collection. Parents and children were not informed of the study prior to participation and thus did not receive compensation for participating.

### 2.2. Intervention Development

An intervention electronic educational flier was distributed to parents and volunteer coaches. Parents and coaches received an email with the flier at two time points during the study—once prior to the start of the league, and once within the first few weeks of the league starting. The email was accompanied with text from the local parks and recreation department that they were “encouraging parent[s] to provide healthier snack and beverage options to teams after games” and to refer to the flier. The flier was also posted on the city’s parks and recreation youth soccer website. The flier was developed by a graphic artist using results from the baseline study [12] that highlighted that sugar consumption, mostly from SSB consumption, was high among youth participants from the baseline study and provided guidance for healthy snacks to bring using adapted Smart Snacks guidelines [21] for sports. The flier also described the number of calories served at an average game, the number of calories expended during an average game, and the average sugar intake each game (through SSB and snack consumption). The flier also gave examples of Smart Snacks to bring to games including fresh fruit, string cheese, popcorn, mixed nuts, and fruit leather. Prior to distribution, the flier was distributed to a small population of parents of school-aged children, and feedback was incorporated based on recommendations from the pilot group. The feedback received from the pilot group suggested changing some of the suggested snack options for more practical options, as well as suggestions for readability and formatting of the flier.

### 2.3. Measurement Instruments and Procedures

Direct observation was used to quantify the occurrence of snacks/beverages offered by parents during or after the game using paper-and-pencil observation forms. Research assistants were trained to record the type, brand, and size of food/beverage offered and the nutritional information including calorie counts and macro/micronutrient contents (sugar, carbohydrates, sodium, cholesterol, fat, saturated fat, fiber, protein) obtained directly from the food label or the manufacturer’s website and record it on paper-and-pencil forms. If homemade products were observed, an approximation of size and type of product was made, and estimations were retrieved from a specific online source (CalorieKing. Available online: www.calorieking.com (accessed on 3 May 2021)). Data were then checked for accuracy by a separated research assistant and were double entered into an electronic database. Research assistants were trained to respond to questions about the study that they were there to observe recreational athletics without describing further details of the observations.

### 2.4. Data Analysis

Data analysis was conducted in SAS 9.4 version (SAS Institute, Cary, NC, USA) using data from the electronic database created from observations. The average of the variable was used as the unit of analysis; variables were summed and divided by the number of games. Univariate analysis was conducted for all nutrition categories (total calories offered, calories for beverages only, calories for snacks only, total sugar, sugar for beverages and snacks only, sodium, carbohydrates, fat, saturated fat, fiber, protein, and cholesterol). Shapiro−Wilk normality tests were conducted on all continuous variables; data were normally distributed and met assumptions. Independent *t*-tests were used to test differences between baseline and comparison groups to determine similarity; additional paired *t*-tests were conducted to test for differences between the baseline and intervention group for caloric intake and macro/micronutrients (alpha level was set at *p* < 0.05). Each macro/micronutrient was averaged for a summation of the amount offered at each game. Because a number of games did not offer snacks or beverages, a separate analysis for these games occurred.

## 3. Results

### 3.1. Participant Characteristics

The study observed 3rd and 4th grade recreational soccer games in two different sports leagues. Observations for the intervention group occurred in 122 games. Observations for the comparison group occurred in 74 games. Baseline data were collected one year prior to the intervention by observing 61 total soccer games in one league.

### 3.2. Nutrition Content

When testing for differences between baseline and comparison data, no significant differences were observed in any of the observed nutrition categories (Table 1), thus indicating that the baseline and comparison groups were similar in terms of nutrition content offered at games. Additionally, the percentage of games that served snacks/beverages did not significantly differ from baseline to comparison groups (data not shown).

Significant differences were observed between the numbers of games that served snacks/beverages from baseline to intervention. During baseline data collection, 80% of games had snacks/beverages brought by parents and after the intervention, just over 50% of games had snacks/beverages brought by parents (*p* < 0.001). (Data not shown.)

Significant differences were also observed in macro/micronutrients offered from baseline to intervention. Specifically, the average amount of total sugar offered from snacks and beverages served at games decreased from 25.5 g per game to 16.7 g per game (*p* < 0.001). Total sugar content offered from SSBs each game also decreased significantly (*p* < 0.001), as did total carbohydrates (*p* < 0.001). Total cholesterol offered each game significantly increased from baseline to intervention. Nonetheless, total calories from all snacks and beverages offered each game did not significantly differ from baseline to intervention (Table 2).

When games not serving snacks and beverages were separately analyzed, total sugar offered from snacks and beverages each game significantly decreased from baseline (28.8 g) to intervention (19.4 g) (*p* < 0.001); significant differences were also observed in total sugar from beverages and total sugar from snacks. Fiber offered at each game significantly increased from baseline to intervention (*p* = 0.01), as did cholesterol (*p* = 0.01). Total calories offered at each game did not significantly differ from baseline to intervention (Table 3).

When examining the types of snacks and beverages offered from baseline to intervention, significant differences were observed among several types of snacks/beverages. The number of times baked goods (e.g., cookies, cakes, donuts) and fruit snacks were offered significantly decreased from baseline to intervention, as well as candy. The number of times snacks including fruits and vegetables, cheese sticks, and granola bars were offered significantly increased from baseline to intervention. The frequency of SSBs offerings (versus not offered) from baseline to intervention significantly decreased, particularly among sports drinks offered. Water offered (versus not offered) significantly increased from baseline to intervention (Table 4).

## 4. Discussion

The present study sought to test the usefulness of a low-cost intervention to decrease calories offered at games through the amount of total sugar offered at games, as well as quantify the snacks/beverages provided by parents through various macro/micronutrients. The major finding from this study was that the amount of total sugar offered each game was reduced in both snacks and beverages after the intervention. Additionally, the snacks and beverages improved through increased offerings of fiber and the number of times fruits, vegetables, and water were offered after the intervention; the number of instances where sports drinks were brought as a beverage by parents was also reduced after the intervention. Lastly, the number of times parents did not bring snacks and beverages to games increased after the intervention.

The American Heart Association recommends that children do not exceed 25 g of added sugar a day [22]. From baseline to postintervention, total sugar offered at games was significantly reduced, yet based on the types of snacks and beverages offered, it was likely that the majority of the total sugars came from added sugar. The reduction occurred in both snacks and beverages. Healthy Eating Research identified public health implications for children and SSB consumption, including risk for diabetes, obesity, and dental caries [23]. Recently published literature has also examined how excess sugar consumption may have a delayed effect on adult obesity [24]; thus, the high rates of sugar consumption among children may have long-term lasting health effects. The American Academy of Pediatrics Committee on Nutrition and Council of Sports Medicine and Fitness states that the average child engaged in physical activity does not need sports drinks [25]. The intervention developed for the present study focused on encouraging parents to bring water; subsequently, water offerings increased after the intervention.

The intervention developed for the present study was also effective at increasing the number of fruits and vegetables provided at youth sporting events across the observation period. Research has indicated that fruit and vegetable consumption remains low among children; daily average total fruit intake is about one cup per day and average vegetable intake is 1.5 cups per day [26]. For children between ages 8–9, recommendations for fruit and vegetable intake are between an average of 1.5 cups to 3 cups of vegetables per day and 1 to 2 cups of fruit per day, depending on age and sex [27]. While this study did not specifically target fruit and vegetable intake, findings from the present study indicated that fruit and vegetable access could be increased in the youth sports setting through intervention. Several multipronged obesity prevention programs have aimed to increase both fruit and vegetable intake while simultaneously decreasing SSB consumption [28]; the present study adds to the literature in this area.

While we did not observe a significant difference in calories offered from baseline to intervention, some of the macronutrient content in the food significantly improved (e.g., decreased total sugar, carbohydrates, and increased fiber). Added sugar contributes to calories without providing essential nutrients; thus, excessive consumption of these may displace more nutrient-dense foods [29].

The present study observed significant increases in the number of dairy products offered from baseline to postintervention, and cholesterol increased after the intervention. While the intervention did not target cholesterol and/or dairy products specifically, string cheese was highlighted as a SmartSnack in the flier distributed to parents; therefore, it was unsurprising that there was an increase in the number of dairy products served at games. Research is mixed about the amount and consumption of dairy for children [30], but it is currently included in the US Dietary Guidelines that children in the age range of the study sample should be consuming about 2.5 cups of dairy each day [25]. Thus, providing snacks and beverages, such as milk and cheese, may help children participating in youth sports to achieve the dietary guidelines.

Lastly, the number of games where food and beverages were not offered significantly changed. While the intervention did not intend on eliminating the snack and beverage offerings, this finding may be an indication that some parents took the initiative to not offer snacks or beverages at all as a way of changing culture. Previous qualitative studies indicate that parents feel pressured about providing snacks/beverages, feel conflicted about the types of beverages and amounts to provide, and want further guidance from organizations on what to constitutes a healthy snack [16,17]. An unintended consequence of this intervention was a reduction in the number of items served at games. Literature states average energy expenditure during youth sports is relatively modest compared to the intake of calories [15], and children generally only receive about 20 min of moderate-to-vigorous physical activity in a game [31,32]; thus, the reduction in the number of games serving snacks or beverages may be positive as an obesity prevention strategy.

### Limitations

This was a quasi-experimental study in two communities; thus, it was limited by the biases inherent with these studies, including non-randomization. Because of the community-based design, the original design was contaminated when one community distributed the intervention to the entire sports league, instead of randomizing participants who received the intervention; thus, we needed to select an additional youth sports league to observe for the post-test. This did not allow for us to determine the true effectiveness of the intervention. Additionally, direct observation was used as a method of quantifying nutritional content. However, trainings and reliability checks were implemented to reduce the bias inherent in direction observation studies. Due to the food label requirements that existed during the data collection period, we were not able to separate total sugar from added sugar, and thus, we were not able to quantify the amount of added sugar offered at each game. Additionally, this study did not assess the impact of the intervention several months after the intervention was distributed, and the baseline and intervention groups were not the same participants; thus, further research is warranted to assess the capacity of this small-scale intervention to maintain usefulness and further test effectiveness. Nevertheless, the results from this study demonstrated that a low-cost educational flier can potentially improve the snacks and beverages brought by parents to youth sports games. Future studies could examine various types of intervention strategies to inform which practice results in the most improvement.

## 5. Conclusions

The results from this study indicate the effectiveness of a low-cost intervention to improve the types of snacks and beverages parents bring to youth recreational sports. Public health practitioners as well as parks and recreation administrators can use the results from this study and the intervention developed to improve the snacks and beverages parents bring to youth recreational sports. From a practical standpoint, this intervention was low-cost and could be easily implemented. Additionally, stakeholders may consider designing policy to limit the amount or types of food and beverages served at youth sports games.

## Figures and Tables

**Table 1 ijerph-18-04886-t001:** Differences in nutritional outcomes per game between baseline intervention group and postintervention comparison group.

Variable	Baseline *n* = 61 Mean (SD) ^	Comparison *n* = 74 Mean (SD) ^	*p*-Value *
Total Calories	228.0 (118.0)	217.0 (83.4)	0.56
Calories for Beverages Only	78.1 (52.3)	61.8 (27.1)	0.05
Calories for Snacks only	169.1 (100.5)	169.6 (78.8)	0.94
Total Sugar (g)	28.8 (13.5)	25.7 (10.4)	0.17
Total Sugar for beverages only	19.3 (12.2)	15.3 (7.0)	0.07
Total Sugar for snacks only	13.0 (11.0)	13 (8.1)	0.99
Sodium (mg)	227.4 (118.0)	186.8 (116.6)	0.05
Carbohydrate (g)	43.0 (19.5)	38.6 (14.2)	0.13
Fat (g)	6.1 (5.3)	6.6 (4.0)	0.53
Saturated Fat (g)	2.0 (2.1)	3.2 (4.3)	0.05
Fiber (g)	0.8 (1.0)	0.6 (0.5)	0.13
Protein (g)	2.5 (5.9)	1.6 (1.1)	0.20
Cholesterol (mg)	1.9 (6.5)	2.9 (6.2)	0.36

* *p*-value < 0.05 indicates significance comparing baseline to comparison; there were no significant differences between these two groups. ^ SD stands for Standard Deviation. *t*-tests were conducted to compare groups.

**Table 2 ijerph-18-04886-t002:** Comparisons between nutritional outcomes per game between baseline intervention to postintervention group.

Variable	Baseline *n* = 61 Mean (SD) ^	Intervention *n* = 122 Mean (SD) ^	*p*-Value
Total calories	201.9 (133.0)	169.4 (110.0)	0.08
Calories for Beverages Only	49.8 (53.0)	38.5 (46.1)	0.159
Calories for Snacks only	144.2 (110.7)	131.0 (90.5)	0.39
Total Sugar (g)	25.5 (15.7)	16.7 (11.6)	<0.001 *
Total Sugar for beverages only	14.9 (13.5)	8.4 (9.3)	<0.001 *
Total Sugar for snacks only	10.5 (8.9)	8.2 (8.0)	0.05
Sodium (mg)	201.3 (199.8)	174.9 (170.9)	0.35
Carbohydrate (g)	38.1 (23.0)	28.4 (17.8)	<0.001 *
Fat (g)	5.4 (5.3)	5.6 (5.6)	0.82
Saturated Fat (g)	1.7 (2.1)	1.8 (2.4)	0.78
Fiber (g)	0.7 (1.0)	1.0 (1.0)	0.06
Protein (g)	2.2 (5.6)	2.7 (3.2)	0.44
Cholesterol (mg)	1.7 (6.2)	4.8 (10.4)	0.03 *

* *p*-value < 0.05 indicates significance comparing baseline to intervention. ^ SD stands for Standard Deviation. *t*-tests were conducted to compare groups.

**Table 3 ijerph-18-04886-t003:** Comparisons between nutritional outcomes per game in baseline intervention to postintervention group eliminating games where no snacks/beverages were served.

Variable	Baseline *n* = 54 Mean (SD) ^	Intervention *n* = 105 Mean (SD) ^	*p*-Value
Total calories (kcals) Calories for beverages only ^†^ Calories for snacks only ^†^	228.0 (118.0) 78.1 (52.3) 169.1 (100.5)	196.8 (92.9) 70.0 (40.7) 155.1 (76.97)	0.07 0.36 0.34
Total Sugar (g) Total Sugar for beverages only ^†^ Total Sugar for snacks only ^†^	28.8 (13.5) 19.3 (12.2) 14.14 (10.79)	19.4 (10.1) 15.3 (10.0) 10.71 (6.60)	<0.001 * 0.03 * 0.03 *
Sodium (mg)	227.4 (118.0)	203.2 (167.8)	0.31
Carbohydrate (g)	43.0 (19.5)	33.0 (14.7)	<0.001 *
Fat (g)	6.1 (5.3)	6.5 (5.5)	0.64
Saturated Fat (g)	2.0 (2.1)	2.1 (2.4)	0.78
Fiber (g)	0.8 (1.0)	1.2 (1.0)	0.01 *
Protein (g)	2.5 (5.9)	3.2 (3.3)	0.30
Cholesterol (mg)	1.9 (6.5)	5.6 (11.0)	0.03 *

^†^ Beverage only: *n* = 47 at baseline and 67 at intervention; snack only: *n* = 52 at baseline; 103 at intervention. * *p*-value < 0.05 indicates significance comparing baseline to intervention. ^ SD stands for Standard Deviation. *t*-tests were conducted to compare groups.

**Table 4 ijerph-18-04886-t004:** Types of snacks and beverages offered during the season between baseline intervention to postintervention group.

Snack/Beverage Offered	Baseline *n* = 316 ^+^	Intervention *n* = 215 ^+^	*p*-Value *
	Frequency (%)	Frequency (%)	
Snacks	171 (54.1)	134 (62.5)	0.06
Baked goods (i.e., cookies, cakes, donuts)	50 (29.20)	19 (14.1)	0.02 *
Fruit Snacks	27 (15.8)	14 (10.4)	0.03 *
Crackers	20 (11.7)	14 (10.4)	0.93
Chips	20 (11.7)	16 (11.9)	0.62
Rice Krispies	17 (9.9)	2 (1.5)	0.006 *
Granola Bar	11 (6.4)	26 (19.3)	<0.001*
Candy	8 (4.7)	0 (0)	0.02 *
Popsicle	6 (3.5)	1 (0.7)	0.25
Fruit/Vegetable	6 (3.5)	20 (14.8)	<0.001 *
Popcorn	5 (2.9)	4 (3.0)	0.99
Pizza	1 (0.6)	-	0.99
Cheese stick	-	11 (8.1)	<0.001*
Yogurt	-	4 (3.0)	0.03*
Apple Sauce	-	2 (1.5)	0.16
Ice Cream	-	1 (0.7)	0.41
Beverage Offered	145 (45.9)	81 (37.5)	0.06
Sugar-Sweetened Beverages	127 (87.6)	65 (80.2)	0.02 *
Juice (e.g., Capri Sun, Kool-Aid)	72 (49.7)	46 (56.8)	0.71
Sports drinks	51 (34.5)	19 (23.5)	0.01 *
Soda	4 (2.8)	-	0.15
Non-SSB	18 (12.4)	16 (19.8)	0.43
Water	5 (3.4)	13 (16.0)	0.005 *
Milk	1 (0.7)	3 (3.7)	0.31
100% Fruit Juice	12 (8.3)	-	0.002 *

* *p*-value < 0.05 indicates significance comparing baseline to intervention. *t*-tests were conducted to compare groups. + indicates the number of snacks/beverages offered by parents at games.

## Data Availability

The data presented in this study are available on request from the corresponding author. The data are not publicly available due to a data sharing agreement with the Provo City Parks and Recreation Department.

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
