# Peer review of "An Educational Intervention for Improving the Snacks and Beverages Brought to Youth Sports in the USA"

_ijerph, 2021, doi:10.3390/ijerph18094886_

Round 1

Reviewer 1 Report

This is an intersesting and useful study focussed on the important area of improving nutritional intake around children's sport. I think some aspects of the manuscript need more clarity and I have tried to articulate this as clearly as I can below:

More Major I feel the intervention needs to be described more clearly in terms of exactly who received the flyer and what was contained within it- how many flyers were distibuted and what exactly was said (if anything) to coached and parents- also i think there may be more than just coaches and parents oinvovled? e.g. food vendors at the sports grounds? but am not sure This is important because in conclusion you state that public health practitioners and parks and recreational staff can use the results to inform their practice-which I think they absolutely could- but what was actually done is not clear enough in the manuscript at the moment- the results look great and are clearly presented but I am not clear enough on how the resutls were acheived and this is the thme of a lot of my feedback/points below I am not absolutely sure who is serving snacks here- the parents are mentioned and I am fairly clear they have been observed by researchers doing this- but where you say just over 80% of 'games served snacks' at baseline and 50% post intervention does this mean some sort of vendor at the game? or are we dealing only with parents here? maybe outline all the avenues the snacks are coming from.

NB you mention improiving the food environment and this is surely more than just what the parents pack to take to the game?

Line 66 how is it Social cognitive theory? I think this needs clarifying- from what I am reading the coaches/parents are provided with material in the form of a flyer which is designed to alter behaviour- e.g. this is how many Kcals you use during a game versus how many there are in snack X and this has a potential cognition impact on the reader? it just needs to be clearer somehow for the reader- maybe a very short precis of SCT with a worked example as i've attemtped above.

Baseline data was carried out a year in advance... so was it the same children a year younger ? or some other grade 3/4 children? I am assuming the same kids and hence the 1 year gap but this needs clarifying.

Line 137 when you say specific online resource which/what do you mean? can you reference it here

Line 212 As per the points above- you state here the food environment was improved- how so?

did vendors change what they were offering?

how did this happen?

where they involved in the study?

did they receive the flyers?

or were there complaints/suggestions by parents to improve the provision etc? it is unclear how they got invovled or why/what would have caused the improvemnt in the food environment

Line 223 can you please check this fruit is1 cup mean avg and veg 1.5 cups mean avg- I ask this because in general dietary surveys tend to show a greater consumption of fruit than vegetables amongst children... in the results of the paper you reference it also looks like there is greater fruit intake than veg.

Minor Line 169 you say data not shown- why not?

Line 284 change'potential' to 'potentially'

Author Response

Thank you for taking the time to review our article. We have uploaded a response to your review (and the other reviews).

Reviewer 1

Author Response

I feel the intervention needs to be described more clearly in terms of exactly who received the flyer and what was contained within it- how many flyers were distributed and what exactly was said (if anything) to coached and parents- also I think there may be more than just coaches and parents involved? e.g. food vendors at the sports grounds? but am not sure This is important because in conclusion you state that public health practitioners and parks and recreational staff can use the results to inform their practice-which I think they absolutely could- but what was actually done is not clear enough in the manuscript at the moment- the results look great and are clearly presented but I am not clear enough on how the results were achieved and this is the theme of a lot of my feedback/points below I am not absolutely sure who is serving snacks here- the parents are mentioned and I am fairly clear they have been observed by researchers doing this- but where you say just over 80% of 'games served snacks' at baseline and 50% post intervention does this mean some sort of vendor at the game? or are we dealing only with parents here? maybe outline all the avenues the snacks are coming from.

In order to add further clarity, we have edited these sentences to indicate that the electronic flier (thus a number of fliers wasn’t distributed) was emailed to parents and coaches once at the beginning of the season and once before the season started. The text used by the Parks & Recreation department as part of the email distribution was added to the narrative as well. (Lines 119-124).

There were no food vendors involved. In this particular jurisdiction, soccer games were played where no concession stands were present. We have added this to line 102-103. This study merely focused on parents.

In order to clarify the “80% of games served snacks at baseline and 50% post-intervention…” we have added that this is specifically referring to parents who brought snacks/beverages. (Lines 184-185)

You mention improving the food environment and this is surely more than just what the parents pack to take to the game?

Our study observed only snacks brought by parents. In this city, there are no food vendors at locations, except for at the recreation center (which only houses youth basketball). There is ample evidence to suggest the vending machines and snack stores at recreation centers can be improved.

We added details about this to highlight that our study only examined snacks brought by parents (Line 81-82)

Line 66 how is it Social cognitive theory? I think this needs clarifying- from what I am reading the coaches/parents are provided with material in the form of a flyer which is designed to alter behaviour- e.g. this is how many Kcals you use during a game versus how many there are in snack X and this has a potential cognition impact on the reader? it just needs to be clearer somehow for the reader- maybe a very short precis of SCT with a worked example as I’ve attempted above.

The sentence discussing the social cognitive theory was referring to a study conducted looking at social cognitive theory and fruit and vegetable intake. After reviewing the paper, we have decided to delete this sentence and reference.

Baseline data was carried out a year in advance... so was it the same children a year younger ? or some other grade 3/4 children? I am assuming the same kids and hence the 1 year gap but this needs clarifying.

These may not be the same children. The way youth leagues are structured within the jurisdictions observed did not make it possible to follow-up the same kids and families one year after baseline. We have added further explanation of this in the methods section. We have added this to the limitations section as well.

Line 137 when you say specific online resource which/what do you mean? can you reference it here

We have added the reference for the Calorie King website. (line 139)

Line 212 As per the points above- you state here the food environment was improved- how so

did vendors change what they were offering?

how did this happen?

where they involved in the study?

did they receive the flyers?

or were there complaints/suggestions by parents to improve the provision etc? it is unclear how they got involved or why/what would have caused the improvement in the food environment

We have altered the purpose statement in the introduction to clarify that this study was focused on assessing the changes in snacks/beverages that parents/guardians brought to games.

We have also added clarification on line 224 that the food environment was improved by snacks/beverages brought by parents. We were not interested in evaluating snacks/beverages served as concessions through vendors, in fact, those options do not exist with the city’s jurisdiction (outside of the recreation center, where none of these observations took place).

After re-reading the paper, the authors recognize the ambiguous nature of the word “food environment” (e.g., may include concession stands, vending machines, etc.) and have altered the manuscript to clarify that the purpose and findings of the study were only relative to snacks/beverages brought by parents (Lines 217-225; 297; 299)

We have also altered the title of the paper to avoid the term “food environment”.

Line 223 can you please check this fruit is1 cup mean avg and veg 1.5 cups mean avg- I ask this because in general dietary surveys tend to show a greater consumption of fruit than vegetables amongst children... in the results of the paper you reference it also looks like there is greater fruit intake than veg.

Thanks for the suggestion- we have added “average” into these sentences to clarify that the reference was looking at average intakes of fruit and vegetables among youth. (Line 251-253)

Minor Line 169 you say data not shown- why not?

Data were not shown in order to reduce the number of tables in the manuscript.

Line 284 change 'potential' to 'potentially'

Thanks for catching this. We have made this change. (Line 309)

Reviewer 2 Report

Please, read the attached document. There were some situations I could not understand but I hope that authors can clarify

Title

Reviewer Comments:

As I will be commenting after, I think that the title should situate readers on the actual context of the study (USA), due to the specific characteristics of this parental practice of bringing food to games

Introduction

Reviewer Comments:

Line 32: in USA (implied but not explicit)

Line 44: what are “recreational soccer games”? I think the expression needs clarification and definition. I think, again, that authors take for granted that USA reality of children's sports is comparable to that of the rest of the world, but it is not. In several European countries these games seem not to exist (there is no longer the “street soccer” and, thus, what children play are organized sports, in clubs (more training that recreation). The same happens on line 42 with the expression “youth recreational sports”.

Line 43: “one study found that 62% of families go out to eat after recreational soccer games”: why should this have “an effect on the overall diet in children”? To go out to eat implies, necessarily, worse (or better) children diet?

Line 44/45: again, this happens on a specific (American) context where parents have a high “influence” on the practice of their children.

Lines 45/46: “…between 300-500 calories and snacks are typically grain-based desserts”: I think authors should explain why this is an issue.

Line 47: “SSB”: first time it appears, should appear between parentheses and be preceded by “sugar-sweetened beverages”

Line 47: “;” or “,”?

Line 48: “SSBs”: it seems there is no reason to differentiate this from previous SSB (both seem plural); furthermore, since I’m not Native American (or English) English speaker, I will not argue on how acronyms make plural in English (the rule seems to be to add an “s”, situation that does not happen in PortugueseJ (acronyms do not change whether they represent singular or plural entities

Lines 85-92: sorry, could not understand the idea…; namely, I could not understand what “n” referred to (games, teams, parents?).

Lines 89, 94: why “comparison group” instead of “control group”? Both groups are to be compared, anyway…

Methods

Reviewer Comments:

I have some concern about two aspects:

1) How much time (one year?) has passed from the baseline and the intervention? How can authors be sure that those previous (baseline) results belonged to the same families (which I think is more important that number of games analyzed) and were actual? That is to say: why not use some of the initial games to collect baseline data and, after, make the intervention? Was the intervention conducted with the same families of those baseline results?

2) Linked to 1), and since it is not explained, how are teams and games constituted? Are there fixed (permanent) teams or are they spontaneous? If they are fixed, why observe several games and not games of several families? The main goal was to see stability on change (i.e., to see if families would change their habits permanently) or to see if many families would change? Did you control which teams/families you observed? I believe that, in a game, if there are (we don’t know) 5, 7, 11 children pre team, there will be 7 families/parents providing food…: how did you control (or not) this? Couldn’t some families worsen their behaviours and others improve it, in the same game and, therefore, these results be masked by analyzing games instead of parents/families?

Results

Reviewer Comment:

Sorry again: can’t understand (tables 1 and 2) what authors mean by “baseline”? Shouldn’t data be the same for baseline group (n=61) in both tables? From what I could (not…) understand, there was an initial baseline result, based on 61 games one year before (right?) and, after the intervention, authors observed 74 games where there was no intervention and 122 where there was an intervention; if that’s so, why the differences

Besides, and despite all the controversy that p-values, by themselves, carry on, there are three results of p=.05 that authors did not consider to represent the existence of significant differences: why?

Line 203: Table 4.Types of Snacks and Beverages Offered” has a different tipe/size of letter

Table 4: again, what are these “n” (316 for baseline, 215 for intervention)?

Limitations

Reviewer Comment:

I particularly enjoyed this section, where authors assume frontally some of the limitations of the study.

Author Response

Thank you for taking the time to review our article. We have uploaded a response to your review (and the other reviews).

Reviewer 2

Author Response

I think that the title should situate readers on the actual context of the study (USA), due to the specific characteristics of this parental practice of bringing food to games

This is a great suggestion given the international focus of the journal. We have altered the title to reflect the location.

Line 32: in USA (implied but not explicit)

We have added clarification to data sources when they come from the USA. These appear throughout the manuscript (e.g. line 33; 44; 50)

Line 44: what are “recreational soccer games”? I think the expression needs clarification and definition. I think, again, that authors take for granted that USA reality of children's sports is comparable to that of the rest of the world, but it is not. In several European countries these games seem not to exist (there is no longer the “street soccer” and, thus, what children play are organized sports, in clubs (more training that recreation). The same happens on line 42 with the expression “youth recreational sports”.

We have added a short definition to clarify what “recreational sports” means in the context of the USA. Recreational sports means individuals do not have to try out for the team and focus on enjoyment and development versus competition and travel. (Line 43-45)

Line 43: “one study found that 62% of families go out to eat after recreational soccer games”: why should this have “an effect on the overall diet in children”? To go out to eat implies, necessarily, worse (or better) children diet?

We have added further clarification on the effect of eating out on children’s diet. For example, two studies were cited that indicated that food consumed outside the home are associated with a higher intake of non-core foods and increased caloric intake. (Line 46-47)

Line 44/45: again, this happens on a specific (American) context where parents have a high “influence” on the practice of their children.

We added that the dissertation studies were conducted in the USA to contextualize the setting and included that parents are typically the ones responsible for providing snacks at youth recreational soccer games (Line 48-49)

Lines 45/46: “…between 300-500 calories and snacks are typically grain-based desserts”: I think authors should explain why this is an issue.

Grain-based desserts are the second most common food source that contribute to added sugar in the US. We have added this clarification to the manuscript on lines 50-51.

Line 47: “SSB”: first time it appears, should appear between parentheses and be preceded by “sugar-sweetened beverages”

Great catch! We have included the definition of the acronym to SSBs on line 52

Line 47: “;” or “,”?

We have changed the semi-colon to a comma on line 53

Line 48: “SSBs”: it seems there is no reason to differentiate this from previous SSB (both seem plural); furthermore, since I’m not Native American (or English) English speaker, I will not argue on how acronyms make plural in English (the rule seems to be to add an “s”, situation that does not happen in PortugueseJ (acronyms do not change whether they represent singular or plural entities

The authors feel it is fairly common in English to add a “s” to pluralize an acronym, but given the international audience, we have changed situations of using SSBs to “SSB consumption”  (e.g. line 53)

Lines 85-92: sorry, could not understand the idea…; namely, I could not understand what “n” referred to (games, teams, parents?).

We have added that the “n” is referring to the number of games observed (Line 92-93)

Lines 89, 94: why “comparison group” instead of “control group”? Both groups are to be compared, anyway…

We have chosen to use the word “comparison group” over “control group” because our design was non-experimental. Teams were not randomized to be part of the intervention or comparison group, thus it was not appropriate to use the term control. (see: https://edpolicy.education.jhu.edu/glossary/#:~:text=Control%20group%3A%20In%20an%20experiment,is%20called%20a%20comparison%20group.

1) How much time (one year?) has passed from the baseline and the intervention? How can authors be sure that those previous (baseline) results belonged to the same families (which I think is more important that number of games analyzed) and were actual? That is to say: why not use some of the initial games to collect baseline data and, after, make the intervention? Was the intervention conducted with the same families of those baseline results?

One year passed from baseline to intervention. Authors do not believe those that received the intervention were likely the same families. Because of the way the sports leagues were organized (3rd and 4th in one league, 5th and 6th in the next league) and potential non-participation from one year to another, it would not have been possible to directly follow-up with only participants in the baseline study. We recognize this as a limitation and have added this to the limitations section.

The intervention was distributed to all participants in the parks and recreation league (from the pre-Kindergarten league and up), but we did not evaluate the effect of the intervention on any other age group except the 3rd and 4th grade participants.

2) Linked to 1), and since it is not explained, how are teams and games constituted? Are there fixed (permanent) teams or are they spontaneous? If they are fixed, why observe several games and not games of several families? The main goal was to see stability on change (i.e., to see if families would change their habits permanently) or to see if many families would change? Did you control which teams/families you observed? I believe that, in a game, if there are (we don’t know) 5, 7, 11 children pre team, there will be 7 families/parents providing food…: how did you control (or not) this? Couldn’t some families worsen their behaviours and others improve it, in the same game and, therefore, these results be masked by analyzing games instead of parents/families?

We have added to line 92 that these are fixed teams for the entire season that are created by the parks and recreation department.

We recognize the flaws of the study relative to not being able to track specific families and have tried to emphasize these appropriately in the limitations section. Given the minimal contact with the teams and parents (e.g. the research team did not have a roster of participants), plus the challenges that arise with various family structures (e.g. divorced parents, parents with multiple children playing multiple sports), it did not seem possible to track these parents. It is possible that some families worsened behaviors and some improved behaviors, but the overall change was positive, as demonstrated in our results. We believe that overall improvement in dietary behavior can have long-term positive benefits for children.

Sorry again: can’t understand (tables 1 and 2) what authors mean by “baseline”? Shouldn’t data be the same for baseline group (n=61) in both tables? From what I could (not…) understand, there was an initial baseline result, based on 61 games one year before (right?) and, after the intervention, authors observed 74 games where there was no intervention and 122 where there was an intervention; if that’s so, why the differences

There’s a difference in the number of games observed because of the number of teams from each jurisdiction and year (they vary) and the number of game cancellations that occur due to weather.  

Besides, and despite all the controversy that p-values, by themselves, carry on, there are three results of p=.05 that authors did not consider to represent the existence of significant differences: why?

In these situations, the p-value was equal to 0.05, it was not less than 0.05. We used the criterion that it must be less than 0.05 and in these situations, they were not.

Line 203: Table 4. “Types of Snacks and Beverages Offered” has a different type/size of letter

We have made this change to make sure the size of the font is the consistent with the rest of the manuscript.

Table 4: again, what are these “n” (316 for baseline, 215 for intervention)?

We have added a footnote to the table to indicate that these “n”s are referring to the number of snacks/beverages offered by parents at games.

Reviewer 3 Report

The authors report a study devoted to test a small-scale intervention and its ability to decrease mainly calories and sugar intake in children participating to a league. It consisted of a single flier developed and distributed to participants at different times of the league and posted on-line. The authors report significant differences between the baseline and the post-intervention data mainly concerning the total sugar amount and the number of added sugar-beverages. They conclud that the study demonstrated that a low cost and easily implemented intervention even if tested in a small-scale intervention can impact on people, that is parents and children, behavior.

  1. Nevertheless, first of all, this reviewer rises a methodological criticism about the role played by the single flier. In fact, the authors report (line 104) that parents and children were not informed of the study prior to participation. However later (line 124) they report that direct observation was used to quantify snacks/beverages number and quality. Thus, the question is: how could research assistants operate without parents and children knowing? The authors have to well clarify this point in the procedures section. This is a crucial point: if participants became in some way informed of the study, the authors can no longer affirm that their results exclusively depended on the plier.
  2. Result section. Four Tables are too much: Graphical data representation (bar histograms for example) are highly recommended to substitute almost some of the shown Tables, making boring the manuscript.
  3. The Discussion section needs of full reorganization, by this reviewer’s opinion. The authors correctly take in consideration one variable a time: sugar content, water, vegetables, calories, dairy and so on. However, in a Discussion, the data really obtained in the study have to be compared with literature data so that the reader can compare numerical and significant or not results of the study with the literature and not generically made affirmations (reduced, increased; I mean).

Lines 207-225 have to be summarized and repetitions eliminated.

New lines have to be inserted line 221-229

  1. The authors recognize the opportunity of studying the impact of the intervention several months later. Don’t the authors think to the opportunity of testing other fliers differently developed and/or did the authors found limitations in the flier they used. An analysis of this point is highly recommended.

Minor: SSB abbreviation was never specified.

Author Response

Thank you for taking the time to review our article. We have uploaded a response to your review (and the other reviews).

Reviewer 3

Author Response

Nevertheless, first of all, this reviewer rises a methodological criticism about the role played by the single flier. In fact, the authors report (line 104) that parents and children were not informed of the study prior to participation. However later (line 124) they report that direct observation was used to quantify snacks/beverages number and quality. Thus, the question is: how could research assistants operate without parents and children knowing? The authors have to well clarify this point in the procedures section. This is a crucial point: if participants became in some way informed of the study, the authors can no longer affirm that their results exclusively depended on the plier.

Parents and children were not given any sort of consent form to participation in the study. Because this is a publically observable behavior, the IRB did not deem a consent form was necessary. It is possible that parents/children saw research assistants at the game, but RAs were instructed to tell parents/children if they were approached that they were observing recreational athletics but were instructed not to share the details of the study. We have added details to the manuscript relative to the training the RAs received on lines 151-153. We have also indicated in the limitations section that there are inherent biases of direct observational studies, like the reviewer identified, but are confident that the results we’ve demonstrated are indicative of the effectiveness of the flier (e.g. baseline and comparison groups were also not informed of the study but were also directly observed and yet the intervention group saw significant changes).

Four Tables are too much: Graphical data representation (bar histograms for example) are highly recommended to substitute almost some of the shown Tables, making boring the manuscript. 

We looked at changing the tables to histograms but the large mean difference between variables (i.e. 228 vs. 0.8) would make assessing differences difficult in histogram form.

We also considered eliminating one or more of the tables, but each of the 4 tables represents another step in the progression of our analysis and all seem relevant. Trying to combine tables would likely get messy and more confusing for the reader. Since the journal instructions provide no explicit limit to the number of tables, we considered the feedback but felt that keeping all tables is allowed and important.

Discussion needs of full reorganization, by this reviewer’s opinion. The authors correctly take in consideration one variable a time: sugar content, water, vegetables, calories, dairy and so on. However, in a Discussion, the data really obtained in the study have to be compared with literature data so that the reader can compare numerical and significant or not results of the study with the literature and not generically made affirmations (reduced, increased; I mean).

We have reworked the discussion in several ways including adding references and a conclusion section. (Lines 318-32)

Lines 207-225 have to be summarized and repetitions eliminated.

The authors are not clear on where repetitions have been used throughout this section.

New lines have to be inserted line 221-229

The authors are not clear what the reviewer means by “inserting new lines”. If this is an editorial comment, the authors will work with the journal to ensure all editorial requests are fulfilled prior to publishing.

The authors recognize the opportunity of studying the impact of the intervention several months later. Don’t the authors think to the opportunity of testing other fliers differently developed and/or did the authors found limitations in the flier they used. An analysis of this point is highly recommended.

Because the flier went through several rounds of pilot testing with parents of children in this age group, we feel confident in the development of the flier in its ability to communicate the information we were hopeful to communicate. That being said, additional studies are warranted to examine interventions that may be more beneficial to improving snacks and beverages brought by parents. We have added this sentence to the limitations section on line 310.

Minor: SSB abbreviation was never specified.

Thank you. We have made this change on line 52

Round 2

Reviewer 1 Report

I think the authors have resonded well to the reviewers suggestions and overall this is a nice, interesng and useful paper focussed on improving children's diets around sport.

Reviewer 2 Report

Thank you for your efforts to enhance your manuscript, which I think were, generally, achieved.

Reviewer 3 Report

In the light of the changes made by the authors, this reviewer think the revised manuscript now suitable for publication.